

# Toward an integrated approach to crop production and pollination ecology through the application of remote sensing

Bryony K. Willcox[1], Andrew J. Robson[2], Brad G. Howlett[3] and Romina Rader[1]

[1] School of Environmental and Rural Science, University of New England, Armidale, NSW, Australia
[2] Precision Agriculture Research Group, University of New England, Armidale, NSW, Australia
[3] The New Zealand Insitute for Plant and Food Research, Christchurch, New Zealand

## ABSTRACT

Insect pollinators provide an essential ecosystem service by transferring pollen to crops and native vegetation. The extent to which pollinator communities vary both spatially and temporally has important implications for ecology, conservation and agricultural production. However, understanding the complex interactions that determine pollination service provisioning and production measures over space and time has remained a major challenge. Remote sensing technologies (RST), including satellite, airborne and ground based sensors, are effective tools for measuring the spatial and temporal variability of vegetation health, diversity and productivity within natural and modified systems. Yet while there are synergies between remote sensing science, pollination ecology and agricultural production, research communities have only recently begun to actively connect these research areas. Here, we review the utility of RST in advancing crop pollination research and highlight knowledge gaps and future research priorities. We found that RST are currently used across many different research fields to assess changes in plant health and production (agricultural production) and to monitor and evaluate changes in biodiversity across multiple landscape types (ecology and conservation). In crop pollination research, the use of RST are limited and largely restricted to quantifying remnant habitat use by pollinators by ascertaining the proportion of, and/or isolation from, a given land use type or local variable. Synchronization between research fields is essential to better understand the spatial and temporal variability in pollinator dependent crop production. RST enable these applications to be scaled across much larger areas than is possible with field-based methods and will facilitate large scale ecological changes to be detected and monitored. We advocate greater use of RST to better understand interactions between pollination, plant health and yield spatial variation in pollinator dependent crops. This more holistic approach is necessary for decision-makers to improve strategies toward managing multiple land use types and ecosystem services.

Corresponding author
Bryony K. Willcox,
bk_willcox@hotmail.com

## INTRODUCTION

Crop pollination research straddles multiple disciplines and scales as it concerns the efficiency and transport of pollen by wild and managed taxa across modified and natural systems. In agricultural systems, pollination services provided by insects are important to 75% of agricultural crop production, which accounts for about 35% of global food supply (*Klein et al., 2007*). Studies that encompass scales relevant to both pollination ecology (plant/tree, landscape) and crop production (field, sub-field) are challenging to conduct using traditional field-based methods. However, if we are to better understand the complex interactions that occur in crops between pollination and production, the development or incorporation of methods and technologies that can facilitate the integration of multiple scales is required.

Remote sensing is defined as the measurement of an object or phenomenon without making physical contact, through the detection of reflected or emitted electromagnetic energy (*Mulla, 2013*). For the assessment of vegetation, remote sensing provides a non-destructive measure of plant status via a range of sensor types (active or passive), resolutions (spatial, temporal, spectral and radiometric) and platforms (satellite, airborne or ground based) (*Mulla, 2013*). There are multiple benefits for utilizing this technology within pollination research, including providing significant opportunities to connect pollination research with other areas already utilizing this technology (Fig. 1). For example, in ecology and conservation, remotely sensed data has been used in biodiversity monitoring and assessment across most habitat types (*Kerr & Ostrovsky, 2003*; *Pettorelli et al., 2005*, *2011*, *2014*; *De Araujo Barbosa, Atkinson & Dearing, 2015*; *Galbraith, Vierling & Bosque-Pérez, 2015*). Applications have included mapping habitat availability, habitat suitability and distributions of threatened species (*Osborne, Alonso & Bryant, 2001*; *Luoto, Kuussaari & Toivonen, 2002*; *Adhikari, Barik & Upadhaya, 2012*) and pest or invasive species (*Andrew & Ustin, 2009*; *Diao & Wang, 2014*), vegetation structure and monitoring (*Seavy, Viers & Wood, 2009*) and biomass estimations (*Dong et al., 2003*). This information is then used to inform conservation and biodiversity management decisions (*Gillespie et al., 2008*). Remote sensing technologies (RST) enable these applications to be scaled across much larger areas than is possible with field-based methods (*Kerr & Ostrovsky, 2003*). In addition, RST offer a standardized, regular source of data allowing large scale ecological changes to be detected and monitored (*Pettorelli et al., 2014*). For example, at a global ecological scale, remote sensing is assisting with the monitoring and mapping anthropogenic impacts including habitat loss, due to deforestation and land conversion, as well as those induced by climate change such as species distribution changes and growing season length (*Kerr & Ostrovsky, 2003*; *Pettorelli et al., 2005*). RST are also used widely in the agricultural sciences to better understand the spatial and temporal variation in crop vigor and productivity and for the monitoring of abiotic and biotic constraints across a range of scales (*Mahlein et al., 2012*; *Mulla, 2013*). In crop production, specific examples include yield forecasting and mapping (*GopalaPillai & Tian, 1999*; *Baez-Gonzalez et al., 2005*; *Zaman, Schumann & Hostler, 2006*; *Robson, Rahman & Muir, 2017*), water and nutrient monitoring and variable rate applications
| RST platforms (*sensor examples*) | Applications | Examples |
|---|---|---|
| Satellite<br>(*Quickbird, WorldView, SPOT, Landsat, RapidEye*)<br> | Land cover mapping, ecosystem classification and change, natural disaster assessment, yield mapping | Robson, Rahman, & Muir, 2017<br>Joyce et al., 2009<br>Pettorelli et al., 2005<br>Tucker, Townshend, & Goff, 1985 |
| Aerial (drones, UAV's, airplanes)<br>(*camera systems e.g. Microsense, Parrot Sequoia*)<br> | Land cover mapping, ecosystem classification and change, natural disaster assessment, wildlife research, precision agriculture | Shahbazi, Théau, & Ménard, 2014<br>Horton et al., 2017 |
| LIDAR (satellite, aerial or terrestrial)<br>(*2D and 3D laser scanners*)<br> | Vegetation (crop and non-crop) mapping and structure | Bradbury et al., 2005<br>Llorens et al., 2011<br>Lefsky et al., 2002 |
| Proximal (tractor or vehicle mounted, handheld devices)<br> | Yield mapping, crop monitoring | Marino & Alvino, 2014<br>Cunningham & Le Feuvre, 2013<br>Cao et al., 2016 |
| Tracking technology (harmonic radar, radio telemetry, satellite)<br> | Animal movement and distributions | Stutchbury et al., 2009;<br>Osborne et al., 1999;<br>Pasquet et al., 2008;<br>Wikelski et al., 2010 |

**Figure 1** **Examples of remote sensing technologies and derived information currently being utilised for agricultural and ecological applications.** Satellite image credit: Thegreenj (Wikipedia), Aerial image credit: Andrew Robson (University of New England), LIDAR image credit: Dan Wu (University of Queensland), Proximal platform image credit: bdk (Wikimedia Commons), Tracking technology image credit: CSIRO. Literature examples— Satellite: *Robson, Rahman & Muir, 2017*; *Joyce et al., 2009*; *Pettorelli et al., 2005*; *Tucker, Townshend & Goff, 1985*. Aerial: *Shahbazi, Théau & Ménard, 2014*; *Horton et al., 2017*. LIDAR: *Bradbury et al., 2005*; *Llorens et al., 2011*; *Lefsky et al., 2002*. Proximal: *Marino & Alvino, 2014*; *Cunningham & Le Feuvre, 2013*; *Cao et al., 2016*. Tracking technology: *Stutchbury et al., 2009*; *Osborne et al., 1999*; *Pasquet et al., 2008*; *Wikelski et al., 2010*.

(*Zaman, Schumann & Miller, 2005*; *Goffart, Olivier & Frankinet, 2008*; *Hedley & Yule, 2009*; *Tremblay et al., 2009*; *Barker & Sawyer, 2010*), growth monitoring (*Gao et al., 2017*), disease detection (*Zhang et al., 2003*; *Mahlein et al., 2012*; *Salgadoe et al., 2018*), orchard flower assessments (*Horton et al., 2017*; *Wang, Underwood & Walsh, 2018*) and weed control (*Lamb & Brown, 2001*). Many crops can now be managed or monitored at multiple scales from individual plants and within field management zones (*Zhang, Wang & Wang, 2002*; *Mulla, 2013*; *Aggelopooulou et al., 2013*) through to regional, national and even global scales (*Wu et al., 2014*). For example, in apple crops the radiometric productive foliar index was developed from the relationship between tree canopy area, normalized difference vegetation index and fruit per tree to better predict crop loads (production per tree (kilogram)), fruit quality measures (fruit weight (gram) and soluble solids (Brix)) (*Best et al., 2008*), while at broader regional, national and global scales, crop monitoring systems utilize a combination of remote sensing data and field data to gauge estimates of crop production, yield and condition (*Wu et al., 2014*).
Crop pollination research has utilized RST to understand landscape and regional scale habitat use by pollinating insects. Pollinators are highly mobile, and some can travel many kilometers for the purpose of foraging and nesting (*Wikelski et al., 2010*). They are impacted by variations in habitat proximity and availability (*Carvalheiro et al., 2011*), with many having nesting and foraging needs directly dependent on specific habitats and some requiring diverse floral resources for year-round foraging needs (*Winfree, Bartomeus & Cariveau, 2011*; *Kennedy et al., 2013*). In a crop environment, variations to insect pollinator movement across different scales can result in downstream variations to pollination and the quantity and quality of crop yields. For example, at local field scales, pollinators can constantly move into and out of flowering crops (*Mesa et al., 2013*), with temporal distributions being significantly influenced by the surrounding landscape (*Holzschuh et al., 2011*). Although flies and bees have been shown to move distances of several hundred metres between the surrounding environment and flowering crops (*Rader et al., 2011*), studies examining pollinator abundance and efficiency are often limited beyond very local scales (individual plants or groups of neighboring plants within a field). While these studies are important for understanding downstream production of fruit and seed set at the plant or within plant scale, they fail to fully elucidate the mechanisms underlying field level variations in crop yield (*Angadi et al., 2003*). In addition, realized crop yields can also be affected by important interacting and confounding factors that vary within and between fields, including plant reproductive strategies, nutrient and water availability, pest pressure, soil health and profile, light availability, crop density and plant health (*Esparza et al., 2001*; *Bos et al., 2007*; *Lundin et al., 2013*; *Motzke et al., 2015*; *Klein et al., 2015*). These multiple and potentially interacting biotic and abiotic factors may also influence pollination success. RST can already detect the signatures of several of these variables (*Apostol et al., 2003*) and therefore, offers much needed scope to be able to explore these interactions at a range of scales. Such studies are a necessary building block in the development of better tailored management practices that optimize pollination and hence yield outcomes for crop growers.

In this study, we conducted a systematic literature review to evaluate the utility of remote sensing technology in crop pollination research. We identify the current suite of remote sensing tools being used, their mode of integration into pollination research, and the unaddressed future research questions that could be answered using RST in crop pollination research.

## MATERIALS AND METHODS

### Literature Search

A literature search using Scopus and Google Scholar identified the types of remote sensing tools utilized in crop pollination research on July 14, 2016 and January 11, 2018. The search terms included "pollination" or pollinator* and "landscape*" or "spatial" or "land cover" or "aerial image*" or "remote sensing" or "satellite" or "tracking" and "crop" or "tree" and "yield" or "fruit set" or "fruit quality" and "wild bee" or "fly" or "diptera" or "beetle" or "native bee" or "bee" or "syrphid" or "hover fly" or "hymenoptera" or

"ant" or "coleopteran." The reference and citation lists of each article identified were also searched, using these same search terms to source additional research papers. Papers were excluded from the database if crop types were not specified. For example, if papers referred to production systems in a general way such as "agroecosystem" or "agricultural matrix" with no further detail about specific crop type, they were excluded.

For each publication we recorded: author/s, journal, year of publication, location of study, crop type, type of RST used (Fig. 1), the spatial factors reported (e.g., proportion or distance to a given land use). We classified each study according to one of three biome types, tropical, subtropical or temperate. The types of insect pollinators investigated in each paper were recorded and categorized into three main groups: managed honey bees (*Apis mellifera*), wild bees (unmanaged *Apis* bees and other non-*Apis* bees, including solitary and social species) and other wild insects (beetles, non-syrphid and syrphid flies, butterflies and moths). Insect pollinator response variables reported in studies were classified into four categories: species richness (number of different species), species abundance (number of individuals of a single species), functional complementarity (how species complement each other in terms of time of visitation, purpose of visit etc.) and community composition (measure of species richness and abundance combined). Finally, we recorded the metric used to measure pollination success. These were categorized into five groups: yield, fruit set, seed set, fruit quality or other (pollen tube or grain counts).

## RESULTS

### Remote sensing technologies used

We identified a total of 68 journal articles (Table S1) that utilized some form of RST to facilitate crop pollination research across 33 different crops (Figs. S1–S4, including general features of data set). The main forms of RST reported in the literature included sourcing aerial (18%) or satellite imagery (12%), using government produced land cover maps (22%) or GPS units (12%). One study was able to access spatially assessed yield data collected via a harvester mounted yield sensor, while another 12% used base maps provided by software programs such as Google Earth™ and ArcGIS® rather than sourcing other imagery. Two papers in the database reported using the outputs from RST but no mention was made of how they obtained this information. The remaining 21% of papers used combinations of these technologies. The most common use for the imagery sourced was to produce land cover maps (79%), using software programs such as ArcGIS®, ENVI® or Google Earth™.

### Methods for incorporating RST

Two methods for incorporating RST into crop pollination were apparent in the literature, broadly defined as proportion and isolation (Fig. 2). Proportion involved creating a set perimeter (i.e., radius of 500 m, one km or two kms) around the focal crop site and then using the proportion of one or more land-use types within that perimeter to explain variations in insect spatial patterns, pollination services or pollination success.
A

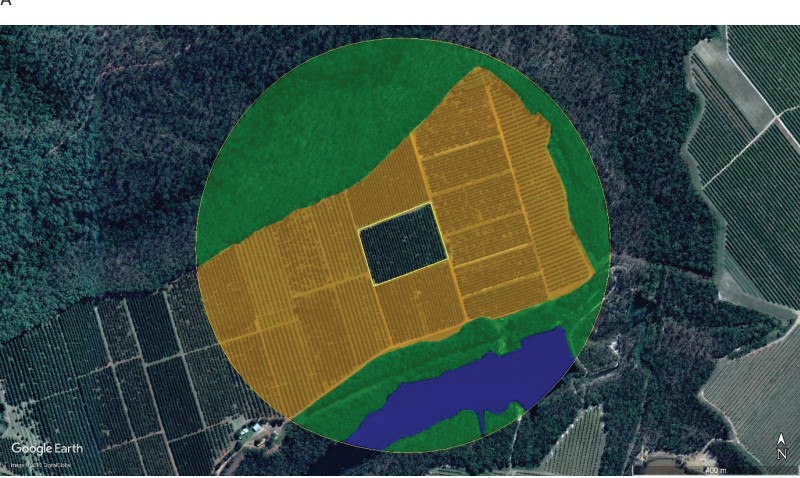

B

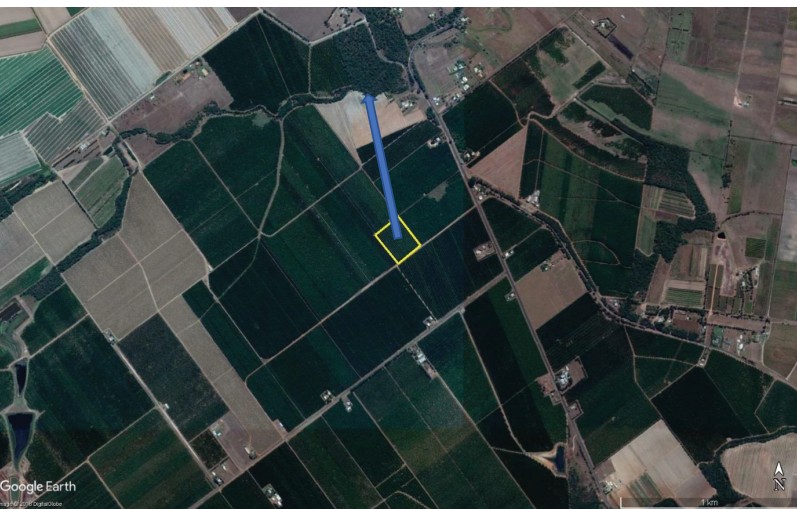

**Figure 2 Examples of proportion and isolation methods.** (A) Proportion of non-crop vegetation (green), crop vegetation (orange) and water (blue) within a 500 m perimeter radius of a focal crop site. (B) Distance of non-crop vegetation from a focal crop site. Map data ©2018 Google, DigitalGlobe.

Landscape types commonly used included non-crop/semi-natural habitat and agricultural/crop. This was the most commonly adopted method across studies (51%).

Isolation was used by 12% of studies. This method measured isolation of a specific crop site or point within a crop site, from landscape or local scale variables, such as natural vegetation patches, natural vegetation strips, natural vegetation margins or honey bee hives. A total of 31% of studies used both proportion and isolation when investigating pollination-related spatial patterns in crops. The remaining 6% of studies assessed spatial variability in various ways including combining decoded honey bee (*A. mellifera*) waggle dances with aerial photographs of the study area to generate foraging distribution maps (*Balfour & Ratnieks, 2016*), combining spatial positions within a crop (edge vs. center) with isolation and/or proportion (*Gibbs et al., 2016*) or assessing field-level yield variability with distance from hives (*Cunningham & Le Feuvre, 2013*).
## Pollination-related response variables

Most crop pollinator studies investigated the extent to which remnant or non-crop vegetation was associated with species richness and species abundance (49%), species abundance alone (21%) or species richness, species abundance and community composition combined (16%). The remaining 14% of studies used other various combinations of these variables with some also including functional complementarity and one paper used a measurement of honey bee hives/ha as their response variable.

Only one study directly used RST (harvester-mounted equipment that linked harvested crop to the GPS location within the field) to gain a measure of field level yield variability (*Cunningham & Le Feuvre, 2013*). While many other studies (53%) considered a measure of crop production (yield, fruit set, seed set or fruit quality) in their experiments these were most often measures taken from an individual plant or group of plants rather than whole fields, and most (82%) were related to the same spatial gradients used for pollinator assessment.

## DISCUSSION

In the existing crop pollination literature, the primary application of remote sensing technology was predominantly limited to defining the extent of non-crop and/or crop vegetation in agricultural production systems. This was reported through two main activities (1) using landscape assessments to measure the *proportion* of land cover types (predominately non-crop, remnant vegetation) surrounding a crop system; and (2) using landscape assessments to determine the *isolation* of a crop system from non-crop vegetation. These measures were then used as explanatory factors for pollinator related measures (species abundance and/or richness), with a smaller proportion of studies also investigating production measures (yield, fruit or seed set or quality) using these same spatial explanatory factors. Pollination studies utilizing RST in this way have provided invaluable information regarding the influence of landscape and regional scale environmental factors on pollinator abundance and distribution. However, given the use of RST in related research areas such as crop production and ecological disciplines, there is a great deal of potential for its further integration into the field of crop pollination.

Despite the influence of landscape scale vegetation factors on pollinator community composition and foraging behavior, relatively little is known about the features within the vegetation prompting these responses. For example, they may be providing alternative floral resources for adult insects, food sources for the development of larval stages or nesting sites. The current methods adopted within crop pollination research do little to address these questions, as the majority of research uses a coarse level of remotely-derived land cover information (*Hofmann et al., 2017*) with high levels of variation in landscape classifications between studies. In our review, over 30 different categories for broad landscape types were reported. To standardize multi-site, crop, year and study comparisons (*Ne'eman et al., 2010*), more consistent descriptions of the vegetation at landscape and regional levels are needed to better understand the features within the vegetation pollinators are responding to. A range of high resolution RST are already being utilized in other research disciplines to extract vegetation data at these more precise

resolutions. For example, aerial LIDAR (Light detection and ranging) and multispectral data were combined to distinguish between seven different vegetation structures which ranged from grasslands to tree stands of varying height and composition (defined as seven fuel type categories in the study) in Mediterranean forests (*García et al., 2011*) and very high spatial resolution data (WorldView-2 satellite) has been used to identify and distinguish between different tree species in a forest environment (*Immitzer, Atzberger & Koukal, 2012*). Integrating this type of technology into current crop pollination research would provide more detailed information about important environmental variables that may be comparable between and within crops, years and regions such as vegetation structure, plant diversity and identity, alternative food resources and nesting habitat. This information would also further inform attempts to model and map pollinator species distributions and pollination services (*Lonsdorf et al., 2009*; *Polce et al., 2013*).

Measuring the impact of the spatial and temporal fluctuations in insect pollinator communities in terms of pollination service provisioning and crop production is still limited for crop pollination research. For example, pollination response variables (visitation, pollen deposition, fruit set and yield) are typically examined at a single plant or row level within a crop field, and while the logistics of conducting these localized experiments often prevent them being conducted at larger scales, they provide little information about the impact of pollination service provisioning at whole field scales. On the other hand, some studies utilize sub-sampling to gain measures that are representative of the whole field (*Goodwin et al., 2011*), while this may provide more indicative field level results it still holds a degree of coarseness. RST can greatly overcome this coarseness, by providing a far more directed method for deciding where to sample within a field, ultimately saving time and effort. *Cunningham & Le Feuvre (2013)* provide one example of how RST can be incorporated to provide field level assessments of pollination service provisioning. Pollination surveys and flower assessments across a whole crop field, designed as a gradient of distance from managed honey bee hives, were used in conjunction with field level yield maps, derived using harvester mounted RST equipment, to determine the effect of managed honey bee density and pollination on yield.

Pollination studies also typically treat crop fields as a single uniform factor in all respects aside from their isolation from or proportion of surrounding landscape and local variables, despite crop production research showing that yield variability can differ both between and within fields (*Robson, Rahman & Muir, 2017*). In comparison, precision agricultural studies recognize field level variability of factors such as nutrients, water stress, soil characteristics and plant health, which impact on final yield measures (*Zaman, Schumann & Miller, 2005*; *Hedley & Yule, 2009*; *Barker & Sawyer, 2010*). Pollination services are typically investigated in isolation from these important abiotic factors that influence yield outcomes. However, a small number of studies have investigated interactions in crop species between pollination, water and nutrients in almond (*Klein et al., 2015*), pollination, pesticide and fertilizer in cucumber (*Motzke et al., 2015*) and pollination, light intensity, nitrogen and water stress in cacao (*Groeneveld et al., 2010*). In all three studies, pollination was the most important driver of yield variability, with fruit set being lowest when insect pollinators were excluded.

However, interactions between pollination and reduced water in almond, pollination and weed control in cucumber and pollination intensity and light intensity in cacao indicate that these need to be more readily incorporated into crop pollination studies (*Groeneveld et al., 2010*; *Klein et al., 2015*; *Motzke et al., 2015*). RST provides the opportunity to use fine-scale measurements of abiotic and environmental factors, such as nutrient or water availability, to unravel their possible interactions with pollination success and assess the field-level heterogeneity of these interactions (*Kerr & Ostrovsky, 2003*; *Pettorelli et al., 2005*; *De Araujo Barbosa, Atkinson & Dearing, 2015*; *Galbraith, Vierling & Bosque-Pérez, 2015*). For example, combining factors important to insect pollinators, such as surrounding vegetation, with field surveys specifically targeted to capture the varying levels of tree or plant stress (water, nutrients, light interception) across a whole field, provide significant opportunities to evaluate these interactions and assess their impact on production measures. RST also offers significant opportunities to evaluate whether fine-scale ecological processes that influence pollination success (e.g., pollinator complementarity) are operating on a much broader scales in a landscape.

To achieve this, several challenges still need to be addressed to enable greater integration of remotely-derived data into crop pollination ecology research. First, access to high resolution remotely sensed data can be costly (*Rose et al., 2015*; *Turner et al., 2015*). Many of the studies identified in this review utilized freely available land use maps or images provided by Government departments or publicly available through sources such as Google Earth™. While these have been well utilized for determining isolation and proportion measures across these studies, they provide a relatively coarse spatial, spectral and temporal resolution of remotely-derived information compared to what is now commercially available and required in many disciplines (*Hofmann et al., 2017*). Second, expert knowledge is required to understand how RST may be best utilized and applied in crop pollination research. Factors to be considered in this process include understanding the type of RST needed and what information the associated data can provide. For example, general patterns of pollinator response to satellite-derived land cover and vegetation information may be further refined through the addition of local structural information of vegetation, which can be increasingly accurately estimated using LIDAR (*Lefsky et al., 2002*; *Andrew, Wulder & Nelson, 2014*). In addition, the information provided by RST needs to be validated through some form of ground-truthing (such as ground-based teams of researchers or sensors) as well as being inspected for possible distortions (atmospheric effects, cloud cover, mechanical issues) (*Turner et al., 2003*; *Andrew, Wulder & Nelson, 2014*). An increased collaborative effort at all stages of the experimental process, between remote sensing experts, crop production and pollination researchers as well as growers and industry stakeholders would enable greater transfer of knowledge among disciplines and facilitate applied outcomes for management.

Understanding how far insect crop pollinators move within and among crops and other habitats is another important future research direction. While several technologies exist to track animal movement at various spatial scales, including radio telemetry,

satellite tracking, harmonic radar and radio frequency identification (*Law & Lean, 1999*; *Bonadonna, Lea & Guinet, 2000*; *Bontadina, Schofield & Naef-Daenzer, 2002*; *Godley et al., 2008*; *Cagnacci et al., 2010*; *Thomas, Baker & Fellowes, 2014*), the small size of insects currently limit the use of these types of technologies (*Kissling, Pattemore & Hagen, 2014*). Micro-transmitters have successfully been deployed to track several larger insect species, such as bumble bees, orchid bees and carpenter bees (*Osborne et al., 1999*; *Pasquet et al., 2008*; *Wikelski et al., 2010*; *Hagen, Wikelski & Kissling, 2011*) however technologies for small to medium pollinators, such as wild bees and hoverflies, are limited. As these technologies become more advanced they may soon become a viable option for more detailed investigations of pollinating insect movement distances within crop pollination research.

## CONCLUSION

As our dependency on pollination services increase, predominately through greater cultivation of pollinator-dependent crops (*Aizen et al., 2008*; *Potts et al., 2010*), understanding the spatial and temporal stability of pollination service provisioning is becoming increasingly relevant. Relationships between pollinator community structure (abundance, richness and evenness) and broad landscape factors as well as pollination services and yield at a single tree or plant level are well established. What is missing is data and an approach that elucidates on the role of pollination at the field and broader scales. This will facilitate our understanding of interactions between pollinators, pollination services and other important environmental, abiotic and pre- and post-pollination factors, such as plant health, water and nutrients, which affect final harvest measures. The further integration of RST may be one approach to investigating these important interactions and an increased collaborative effort between agricultural and pollination ecology researchers, remote sensing experts as well as key industry stakeholders such as growers will help ensure outcomes that are more applicable for managing multiple land use types and ecosystem services.

### Funding

Bryony K. Willcox was supported by a PhD scholarship from the University of New England. Bryony K. Willcox and Andrew J. Robson were funded by RnD4Profit-14-01-008 "Multi-scale monitoring tools for managing Australian Tree Crops: Industry meets innovation." Romina Rader was supported by the Ian Potter Foundation (ref:20160225), a Rural Industries Research and Development Corporation grant for the project "Secure Pollination for More Productive Agriculture (RnD4Profit-15-02-035)" and an Australian Research Council Discovery Early Career Researcher Award DE170101349. Brad G. Howlett and Romina Rader were supported by the Ministry of Business, Innovation and Employment C11X1309. The funders had no role in study design, data collection and analysis, decision to publish, or preparation of the manuscript.

## Grant Disclosures

The following grant information was disclosed by the authors:

PhD scholarship from the University of New England.

RnD4Profit-14-01-008 "Multi-scale monitoring tools for managing Australian Tree Crops: Industry meets innovation".

Ian Potter Foundation (ref:20160225), a Rural Industries Research and Development Corporation grant for the project "Secure Pollination for More Productive Agriculture": RnD4Profit-15-02-035.

Australian Research Council Discovery Early Career Researcher Award: DE170101349.

Ministry of Business, Innovation and Employment: C11X1309.

## Competing Interests

Brad G. Howlett is employed by The New Zealand Institute for Plant and Food Research Limited.

## Author Contributions

- Bryony K. Willcox prepared figures and/or tables, authored or reviewed drafts of the paper, approved the final draft.
- Andrew J. Robson authored or reviewed drafts of the paper, approved the final draft.
- Brad G. Howlett authored or reviewed drafts of the paper, approved the final draft.
- Romina Rader authored or reviewed drafts of the paper, approved the final draft.

## Data Availability

The research in this article did not generate any data or code. As a review article no data was generated.

## Supplemental Information

Supplemental information for this article can be found online at http://dx.doi.org/10.7717/peerj.5806#supplemental-information.

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
