# Peer review of "Toward an integrated approach to crop production and pollination ecology through the application of remote sensing"

_PeerJ, doi:10.7717/peerj.5806_

## Round 0.1 · original submission · Major Revisions

· Academic Editor

Major Revisions

Thank you for submitting your manuscript to PeerJ. The reviewers have identified major revisions to your manuscript. For example: 1) the addition of urban pollination RST references, 2) moving RST technology brief explanations up into introduction/background to accompany Figure 1 and expand for clarity versus in discussion, 3) expand material addressing RST other than satellite imagery, and, 4) use data to support your argument of spurious upscaling made by most pollination studies, and show how RST could contribute to it. We look forward to recievimg your revised manuscript.

Reviewer 1 ·

Basic reporting

Introduction is clear, concise, highly relevant, and of cross-disciplinary interest.
Suggest adding several RST studies on pollination in urban settings, such as urban farms, community gardens, and green roofs to round out manuscript.
Suggest moving RST technology brief explanations up into introduction/background to accompany Figure 1 and expand for clarity versus in discussion section.

Experimental design

As suggested above, urban context appears to be missing. This will enhance the manuscript.

Validity of the findings

Suggest moving RST technology explanations up into introduction and expand, for reader clarity and orientation.
Lines 275 p 278: suggest expanding on the need for interdisciplinary, transdisciplinary benefits for academia as well as lay practitioners and other key stakeholders. This is an important point that cannot be overemphasized, as it is a current gap as well as future research opportunity.
Conclusion will benefit from further strengthening of final key points and the argument for RST.
Some of the study data (i.e. % of papers said X) might work well in a table for better

Additional comments

Much needed design paper for incorporating RST tools and methodologies into pollination services research.

Reviewer 2 ·

Basic reporting

The paper is well written and well structured. The review is broad and raises some good points, but it is mainly a cursory survey of the literature.

Experimental design

The literature review was logically conducted, easily repeatable and the paper was well structured and easy to read.

Validity of the findings

The paper read as an introduction to something bigger rather than a stand-alone piece. Specifically, although the authors mention five types of remote sensing technologies (RST) that could be applied to pollination ecology and crop production, but the bulk of the paper was about the use of satellite imagery. The other RSTs were mentioned only briefly and it was not clear why such technologies would answer questions in the field better than current methods.
While I appreciated the call for "increased collaborative effort between remote sensing experts, crop production and pollination researchers" (lines 276-277), the paper would benefit from an expansion of this point. Such expansion will also likely aid in addressing how the mentioned RSTs would be better than current methods (see above).

Reviewer 3 ·

Basic reporting

no comment

Experimental design

no comment

Validity of the findings

no comment

Additional comments

General comments

This manuscript presents the results of a literature review about how pollination research integrate remote sensing technologies (RST) to measure environmental variables that might influence pollinator and pollination or use RST to assess pollination success (yield). The authors discuss this results in terms of research potential that could result from a better integration of RST in future pollination research.

The article is nicely written and it is actually really interesting to have a summary about the use of RST in pollination studies. There is a need for the integration of environmental variables in pollination studies that actually match the ecology of insect pollinators, going beyond variables like “proportion of semi-natural habitats” (not all habitats are good for pollinators, and some patches of the same habitat types vary in the resource quality they provide). The authors managed to explain why this is important to measure and how to measure this.
However, I am less convinced by the part of the article dealing with fine-scale measurement of yields and environmental factors that can affect plant growth. The authors state that most of the pollination studies should not extrapolate their result at the field scale from measurement made at the plant scale or at the scale of a group of plants (quadrat or crop row for example), because there are many variations within crop fields that influence yield production such as water availability, soil quality and sun interception (for example). Several problems arise from this section:
- The authors did not explore the relative benefit of measuring pollination on the whole field versus on some plant individuals. How the field-level yield is different from the yield measured at the scale of quadrats or plants? If the study design is well established, the quadrats or individual plants should be selected to be representative of the whole field. A better study design or an increase in sampling effort could be enough to overcome the within-field heterogeneity
- The author do not mention that most of the studies exploring the effect of pollinator community on pollination are not primarily interested in this fine-scale abiotic factors that could influence the yield. Many studies actually use some techniques to avoid such confounding effects (comparing bagged versus open flowers from the same locations, using potted plants with the same soil and water availability, increasing sample size to remove possible correlation of those abiotic factors and the variables of interest, etc.). In the field of precision agriculture, I think that measuring those fine-scale abiotic factors and their impact on yield is really important, but I am not sure that those fine scale mechanisms actually put into questions the results of studies finding an “added” value of pollination on yields.
- The authors recommend to use fine-scale measurement of abiotic and environmental factors using RST (such as nutrient or water availability) to unravel possible interactions between pollination success and those factors. However, the authors state in the text that field-based pollination assessment are not currently able to capture the within-field heterogeneity in pollination success. How to solve this problem? Can we scale-up pollination success measured on individuals or groups of individuals on large spatial scale? Are the studies exploring fine-scale ecological processes acting on pollination success such as species complementarity still valid at the field scale or at larger spatial scale? Could RST contribute to this upscaling? I think that the authors have not tackled those questions in the first place.

The authors should consider using some data to support their argument of spurious upscaling made by most of pollination studies, and show how RST could contribute to it.

Specific comments

Introduction

L 90-91: Andersson et al 2014 is not about pollinating insects but about pollination success of field bean. You might use another reference to illustrate your idea.
L 92-93: The first part of the sentence should be improved: ”habitat needs …dependent on specific habitats”.
L 95: ”variation to” --> ”variation in”
L 98: ” bee pollinators” --> ”bees”
L 98-99: You should rewrite this sentence because it gives the feeling that bees actually disperse from flowering crops, into the wider landscape. They only use them as food resources, and nest in some other habitat patches.
L 103-108: Some cited studies do no deal with pollination success but with pollinator communities (sampled in some habitats), so it is rather strange to use them to illustrate the extrapolation of results from plant scale to larger scales. And some article do not explicitely extrapolate their results: Holzschuh et al 2007 measured some explanatory variables at the landscape scale but still draw their conclusion at the field scale (scale of measurement of the response variable).

Discussion
L 220-221: ”precise scale” --> ”precise resolution”
L 232: ”these spatial and temporal fluctuations” --> ”the spatial and temporal fluctuations” because the previous paragraph refers to environmental variables measured through remote sensing tools, not to remote sensing tools used to infer pollinator community structure and pollination success.

---

## Round 0.2 · accepted · Accept

· Academic Editor

Accept

Thank you for addressing the many points raised by the reviewers. Your manuscript is now ready for publication in PeerJ.

# Reviewer 2 ·

Basic reporting

No Comment

Experimental design

No Comment

Validity of the findings

No Comment

Additional comments

The authors have adequately addressed my issues with the initial version and this paper is now ready to be accepted for publication.